# Distributed Differential Privacy in Multi-Armed Bandits

**Sayak Ray Chowdhury**[*]
Microsoft Research, India
t-sayakr@microsoft.com

**Xingyu Zhou**[*]
Wayne State University, USA
xingyu.zhou@wayne.edu

## Abstract

We consider the standard $K$-armed bandit problem under a distributed trust model of differential privacy (DP), which enables to guarantee privacy without a trustworthy server. Under this trust model, previous work largely focus on achieving privacy using a shuffle protocol, where a batch of users data are randomly permuted before sending to a central server. This protocol achieves $(\varepsilon, \delta)$ or approximate-DP guarantee by sacrificing an additive $O\left(\frac{K \log T \sqrt{\log(1/\delta)}}{\varepsilon}\right)$ factor in $T$-step cumulative regret. In contrast, the optimal privacy cost to achieve a stronger $(\varepsilon, 0)$ or pure-DP guarantee under the widely used central trust model is only $\Theta\left(\frac{K \log T}{\varepsilon}\right)$, where, however, a trusted server is required. In this work, we aim to obtain a pure-DP guarantee under distributed trust model while sacrificing no more regret than that under central trust model. We achieve this by designing a generic bandit algorithm based on successive arm elimination, where privacy is guaranteed by corrupting rewards with an equivalent discrete Laplace noise ensured by a secure computation protocol. We numerically simulate regret performance of our algorithm, which corroborates our theoretical findings.

## 1 Introduction

The multi-armed bandit (MAB) [1] problem provides a simple but powerful framework for sequential decision-making under uncertainty with bandit feedback, which has attracted a wide range of practical applications such as online advertising [2], product recommendations [3], clinical trials [4], to name a few. Along with its broad applicability, however, there is an increasing concern of privacy risk in MAB due to its intrinsic dependence on users' feedback, which could leak users' sensitive information [5].

To alleviate the above concern, the notion of *differential privacy*, introduced by Dwork et al. [6] in the field of computer science theory, has recently been adopted to design privacy-preserving bandit algorithms (see, e.g., [7–9]). Differential privacy (DP) provides a principled way to mathematically prove privacy guarantees against adversaries with arbitrary auxiliary information about users. To achieve this, a differentially private bandit algorithm typically relies on a well-tuned random noise to obscure each user's contribution to the output, depending on privacy levels $\varepsilon, \delta$ – smaller values lead to stronger protection but also suffer worse utility (i.e., regret). For example, the central server of a recommendation system can use random noise to perturb its statistics on each item after receiving feedback (i.e., clicks/ratings) from users. This is often termed as *central model* [10], since the central server has the trust of its users and hence has a direct access to their raw data. Under this model, an optimal private MAB algorithm with a pure DP guarantee (i.e., when $\delta = 0$) is proposed in [11], which only incurs an *additive* $O\left(\frac{K \log T}{\varepsilon}\right)$ term in the cumulative regret compared to the standard setting when privacy is not sought after [12]. However, this high trust model is not always feasible in practice since users may not be willing to share their raw data directly to the server. This motivates

---

[*]Equal contributions

2022 Trustworthy and Socially Responsible Machine Learning (TSRML 2022) co-located with NeurIPS 2022.

Table 1: Best-known performance of private MAB under different privacy models ($K$ = number of arms, $T$ = time horizon, $\Delta_a$= reward gap of arm $a$ w.r.t. best arm, $\varepsilon, \delta, \alpha$ = privacy parameters)

| Trust Model | Privacy Guarantee | Best-Known Regret Bounds |
|---|---|---|
| Central | $(\varepsilon, 0)$-DP | $\Theta\left(\sum_{a\in[K]:\Delta_a>0} \frac{\log T}{\Delta_a} + \frac{K\log T}{\varepsilon}\right)$ [11, 9] |
| Local | $(\varepsilon, 0)$-DP | $\Theta\left(\frac{1}{\varepsilon^2}\sum_{a\in[K]:\Delta_a>0} \frac{\log T}{\Delta_a}\right)$ [14] |
| Distributed | $(\varepsilon, \delta)$-DP | $O\left(\sum_{a\in[K]:\Delta_a>0} \frac{\log T}{\Delta_a} + \frac{K\log T\sqrt{\log(1/\delta)}}{\varepsilon}\right)$ [22] |
| Distributed | $(\varepsilon, 0)$-DP | $O\left(\sum_{a\in[K]:\Delta_a>0} \frac{\log T}{\Delta_a} + \frac{K\log T}{\varepsilon}\right)$ (Thm. 1) |

to employ a *local model* [13] of trust, where DP is achieved without a trusted server as each user perturbs her data prior to sharing with the server. This ensures a stronger privacy protection, but leads to a high cost in utility due to large aggregated noise from all users. As shown in [14], under the local model, private MAB algorithms have to incur a *multiplicative* $1/\varepsilon^2$ factor in the regret rather than the additive one in the central model.

In attempts to recover the same utility of central model without a trustworthy server like the local model, an intermediate DP trust model called *distributed model* has gained an increasing interest, especially in the context of (federated) supervised learning [15–19]. Under this model, each user first perturbs her data via a local randomizer, and then sends the randomized data to a *secure* computation function. This secure function can be leveraged to guarantee privacy through aggregated noise from distributed users. There are two popular secure computation functions: *secure aggregation* (SecAgg) [20] and *secure shuffling* [21]. The former often relies on cryptographic primitives to securely aggregate users' data so that the server only learns the aggregated result, while the latter securely shuffle users' messages to hide their source. To the best of our knowledge, distributed DP model is far less studied in online learning as compared to supervised learning, with only known results for standard $K$-armed bandits in [22] where only secure shuffling is adopted. Despite being pioneer work, the results obtained in this paper have several limitations: (i) The privacy guarantee is obtained only for approximate DP ($\delta > 0$) – a stronger pure DP ($\delta = 0$) guarantee is not achieved; (ii) The cost of privacy is a multiplicative $\sqrt{\log(1/\delta)}$ factor away from that of central model, leading to a higher regret bound; (iii) The secure protocol works only for binary rewards (or communication intensive for real rewards). All of these lead to the following primary question:

*Is there a communication-efficient MAB algorithm that satisfies pure DP in the distributed model while attaining the same regret bound as in the central model?*

**Our contributions.** We answer this in the affirmative (see Table 1) by overcoming several key challenges that arise in the distributed DP model for bandits. First, although secure aggregation protocol offers a benefit in communication cost, it works only in the integer domain due to an inherent modular operation [20]. This immediately requires substantial changes to existing private bandit algorithms, including data quantization, distributed *discrete privacy noise* and modular summation arithmetic. Second, compared to supervised learning where typically a bound on the noise variance is sufficient to analyse utility, regret analysis of private bandits require a tight tail bound. This gets challenging due to aforementioned requirements and communication constraints in the distributed model. We take a systematic approach to address these challenges, which is summarized below:

**1.** We propose a private bandit algorithm using a batch-variant of the successive arm elimination technique as a building block. We ensure distributed DP via a private protocol $\mathcal{P} = (\mathcal{R}, \mathcal{S}, \mathcal{A})$ tailored to discrete noise and modular operation (see Algorithm 1). It consists of a local randomizer $\mathcal{R}$ at each user, a generic secure computation function $\mathcal{S}$, and an analyzer $\mathcal{A}$ at the server. This template protocol enables us to achieve different privacy guarantees by tuning the noise in $\mathcal{R}$.

**2.** To achieve pure DP guarantee, we instantiate $\mathcal{R}$ at each user with an appropriate Pólya random noise so that the total noise seen by the server is a discrete Laplace. Using tail properties of discrete Laplace, we show that the cumulative regret of our algorithm matches the one in the central model, achieving the minimax rate under pure DP (see Theorem 1). Moreover, the communication bits per-user scale only logarithmicaly with the number of participating users in each batch. We numerically evaluate the regret performance of our algorithm, which corroborate our theoretical results.

## 2 Preliminaries

In this section, we formally introduce the distributed differential privacy model in bandits. Before that we recall the learning paradigm in multi-armed bandits and basic differential privacy definitions.

**Learning Model and Regret in MAB.** At each time slot $t \in [T] := \{1, \ldots, T\}$, the agent (e.g., recommender system) selects an arm $a \in [K]$ (e.g., an advertisement) and obtains an i.i.d reward $r_t$ from user $t$ (e.g., a rating indicating how much she likes it), which is sampled from a distribution over $[0, 1]$ with mean given by $\mu_a$. Let $a^* := \mathrm{argmax}_{a \in [K]} \mu_a$ be the arm with the highest mean and denote $\mu^* := \mu_{a^*}$ for simplicity. Let $\Delta_a := \mu^* - \mu_a$ be the gap of the expected reward between the optimal arm $a^*$ and any other arm $a$. Further, let $N_a(t)$ be the total number of times that arm $a$ has been played in the first $t$ rounds. The goal of the agent is to maximize its total reward, or equivalently to minimize the cumulative expected pseudo-regret, defined as

$$\mathbb{E}\left[\mathrm{Reg}(T)\right] := T \cdot \mu^* - \mathbb{E}\left[\sum_{t=1}^{T} r_t\right] = \mathbb{E}\left[\sum_{a \in [K]} \Delta_a N_a(T)\right].$$

**Differential Privacy.** Let $\mathcal{D} = [0, 1]$ be the data universe, and $n \in \mathbb{N}$ be the number of *unique* users. we say $D, D' \in \mathcal{D}^n$ are neighboring datasets if they only differ in one user's reward $D_i$ for some $i \in [n]$. With this, we have the following standard definition of differential privacy [6].

**Definition 1** (Differential Privacy). *For $\varepsilon, \delta > 0$, a randomized mechanism $\mathcal{M}$ satisfies $(\varepsilon, \delta)$-DP if for all neighboring datasets $D, D'$ and all events $\mathcal{E}$ in the range of $\mathcal{M}$, we have*

$$\mathbb{P}\left[\mathcal{M}(D) \in \mathcal{E}\right] \le e^\varepsilon \cdot \mathbb{P}\left[\mathcal{M}(D') \in \mathcal{E}\right] + \delta.$$

The special case of $(\varepsilon, 0)$-DP is often referred to as *pure differential privacy*, whereas, for $\delta > 0$, $(\varepsilon, \delta)$-DP is referred to as *approximate differential privacy*.

**Distributed Differential Privacy.** A distributed bandit learning protocol $\mathcal{P} = (\mathcal{R}, \mathcal{S}, \mathcal{A})$ consists of three parts: (i) a (local) randomizer $\mathcal{R}$ at each user's side, (ii) an intermediate secure protocol $\mathcal{S}$, and (iii) an analyzer $\mathcal{A}$ at the central server. Each user $i$ first locally apply the randomizer $\mathcal{R}$ on its raw data (i.e., reward) $D_i$, and sends the randomized data to a secure computation protocol $\mathcal{S}$ (e.g., secure aggregation or shuffling). This intermediate secure protocol $\mathcal{S}$ takes a batch of users' randomized data and generates inputs to the central server, which utilizes an analyzer $\mathcal{A}$ to compute the output (e.g., action) using received messages from $\mathcal{S}$.

The secure computation protocol $\mathcal{S}$ has two main variations: *secure shuffling* and *secure aggregation*. Both of them essentially work with a batch of users' randomized data and guarantee that the central server cannot infer any individual's data while the total noise in the inputs to the analyzer provides a high privacy level. To adapt both into our MAB protocol, it is natural to divide participating users into batches. For each batch $b \in [B]$ with $n_b$ users, the outputs of $\mathcal{S}$ is given by $\mathcal{S} \circ \mathcal{R}^{n_b}(D) := \mathcal{S}(\mathcal{R}(D_1), \ldots, \mathcal{R}(D_{n_b}))$. The goal is to guarantee that the the view of all $B$ batches' outputs satisfy DP. To this end, we define a (composite) mechanism

$$\mathcal{M}_\mathcal{P} = (\mathcal{S} \circ \mathcal{R}^{n_1}, \ldots, \mathcal{S} \circ \mathcal{R}^{n_B}),$$

where each individual mechanism $\mathcal{S} \circ \mathcal{R}^{n_b}$ operates on $n_b$ users' rewards, i.e., on a dataset from $\mathcal{D}^{n_b}$. With this notation, we have the following definition of distributed differential privacy.

**Definition 2** (Distributed DP). *A protocol $\mathcal{P} = (\mathcal{R}, \mathcal{S}, \mathcal{A})$ is said to satisfy DP in the distributed model if the mechanism $\mathcal{M}_\mathcal{P}$ satisfies Definition 1.*

In the central DP model, the privacy burden lies with a central server (in particular, analyzer $\mathcal{A}$), which needs to inject necessary random noise to achieve privacy. On the other hand, in the local DP model, each user's data is privatized by local randomizer $\mathcal{R}$. In contrast, in the distributed DP model, privacy without a trusted central server is achieved by ensuring that the inputs to the analyzer $\mathcal{A}$ already satisfy differential privacy. Specifically, by properly designing the intermediate protocol $\mathcal{S}$ and the noise level in the randomizer $\mathcal{R}$, one can ensure that the final added noise in the aggregated data over a batch of users matches the noise that would have otherwise been added in the central model by the trusted server. Through this, distributed DP model provides the possibility to achieve the same level of utility as the central model without a trustworthy central server.

## 3 A Generic Algorithm for Private Bandits

In this section, we propose a generic algorithmic framework for multi-armed bandits under the distributed privacy model.

**Batch-Based Successive Elimination Algorithm.** Our generic algorithm (Algorithm 1) builds upon the classic idea of successive arm elimination [23] with the additional incorporation of batches and a black-box protocol $\mathcal{P} = (\mathcal{R}, \mathcal{S}, \mathcal{A})$ to achieve distributed differential privacy. More specifically, it divides the time horizon $T$ into batches of exponentially increasing size and eliminates sub-optimal arms successively. To this end, for each active arm $a$ at batch $b$, it first prescribes arm $a$ to a *batch* of $l(b)$ new users.[2] After pulling the prescribed action $a$, each user applies the local randomizer $\mathcal{R}$ to her reward and sends the randomized reward to the intermediary function $\mathcal{S}$, which runs a secure protocol (e.g., secure aggregation or secure shuffling) over the total $l(b)$ number of randomized rewards. Then, upon receiving the outputs of $\mathcal{S}$, the server applies the analyzer $\mathcal{A}$ to compute the the sum of rewards for batch $b$ when pulling arm $a$, which gives the new mean estimate $\widehat{\mu}_a(b)$ of arm $a$ after being divided by the total pulls $l(b)$. Then, upper and lower confidence bounds, $\text{UCB}_a(b)$ and $\text{LCB}_a(b)$, respectively, are computed around the mean estimate $\widehat{\mu}_a(b)$ with a properly chosen confidence width $\beta(b)$. Finally, after the iteration over all active arms in batch $b$ (denoted by the set $\Phi(b)$), it adopts the standard arm elimination criterion to remove all obviously sub-optimal arms, i.e., it removes an arm $a$ from $\Phi(b)$ if $\text{UCB}_a(b)$ falls below $\text{LCB}_{a'}(b)$ of any other arm $a' \in \Phi(b)$. It now remains to design a distributed DP protocol $\mathcal{P}$, which will be explored at length in the next section.

**Distributed DP Protocol via Discrete Privacy Noise.** In this section, inspired by [24, 25], we provide a general template protocol $\mathcal{P}$ for the distributed DP model, which rely only on discrete privacy noise. The motivation behind using discrete noise is three-fold: (i) Practical secure aggregation (SecAgg) functions work only on the integer domain [20]; (ii) A real-value noise is often difficult to encode on finite computers in practice [26, 17] and a naive use of finite precision approximation may lead to a possible failure of privacy protection [27]; (iii) Discrete noise enables communication via bits rather than real numbers, hence reducing communication overheads. The detail of our template protocol $\mathcal{P} = (\mathcal{R}, \mathcal{S}, \mathcal{A})$ for distributed DP model is as follows. The local randomizer $\mathcal{R}$ receives user's real-valued reward and encodes it as an integer via a fixed-point encoding with precision $g > 0$ and randomized rounding. Then, it generates a discrete noise, which depends on the specific privacy-regret trade-off requirement (to be discussed later under specific mechanisms). Next, it adds the random noise with encoded reward, modulo clips the sum and sends the final integer as input to $\mathcal{S}$. We leave $\mathcal{S}$ as a black-box function that implements secure aggregation. Engineering implementations of this function is beyond the scope of this paper. Instead, a high-level idea behind this technique is to ensure that after receiving messages from $\mathcal{S}$, the server cannot distinguish each individual's message. Finally, the job of the analyzer $\mathcal{A}$ in our template protocol is to calculate the sum of rewards within a batch as accurately as possible. To this end, it directly corrects for possible underflow due to modular operation and bias due to encoding by $g$. To sum it up, the end goal of our protocol $\mathcal{P}$ is to ensure that it provides the required privacy protection while guaranteeing an output $z \approx \sum_{i=1}^n x_i$ with high probability, which is the key to our privacy and regret analysis.

## 4 Achieving Pure DP in the Distributed Model

In this section, we show that Algorithm 1 with a specific instantiation of the template protocol $\mathcal{P}$ is able to achieve pure-DP in the distributed DP model via secure aggregation. As mentioned before, we will treat SecAgg as a black-box function, which implements the following procedure: given $n$ users and their randomized messages $y_i \in \mathbb{Z}_m$ (i.e., integer in $\{0, 1, \ldots, m-1\}$) obtained via $\mathcal{R}$, the SecAgg function $\mathcal{S}$ faithfully computes the modular sum of the $n$ messages, that is, $\widehat{y} = (\sum_{i=1}^n y_i) \bmod m$ (i.e., it is perfectly correct), while revealing no further information (e.g., individual message) to a potential attacker (i.e., it is perfectly secure). Now, to guarantee privacy in the distributed model, we need to carefully determine the amount of (discrete) noise in $\mathcal{R}$ so that the total noise in a batch provides $(\varepsilon, 0)$-DP. One natural choice is the discrete Laplace noise.

**Definition 3** (Discrete Laplace Distribution). *Let $b > 0$. A random variable $X$ has a discrete Laplace distribution with scale parameter $b$, denoted by $\textbf{Lap}_{\mathbb{Z}}(b)$, if it has a probability mass function given by*

$$\forall x \in \mathbb{Z}, \quad \mathbb{P}[X = x] = \frac{e^{1/b} - 1}{e^{1/b} + 1} \cdot e^{-|x|/b}.$$

A key property of discrete Laplace that we will use is its *infinite divisibility*, which allows us to simulate it in a distributed way [28, Theorem 5.1].

---

[2]In contrast, the classic successive elimination algorithm prescribes each arm to a single user.

---
**Algorithm 1** Private Batch-Based Successive Arm Elimination
---
1: **Parameters:** # arms $K$, Time horizon $T$, privacy level $\varepsilon > 0$, Confidence radii $\{\beta(b)\}_{b \geq 1}$
2: **Initialize:** Batch count $b = 1$, Active arm set $\Phi(b) = \{1, \ldots, K\}$, Estimate $\widehat{\mu}_a(1) = 0$, $\forall a \in [K]$
3: **for** batch $b = 1, 2, \ldots$ **do**
4:     Set batch size $l(b) = 2^b$
5:     **for** each active arm $a \in \Phi(b)$ **do**
6:         **for** each new user $i$ from 1 to $l(b)$ **do**
7:             Pull arm $a$ and generate reward $r_a^i(b)$
8:             Send randomized data $y_a^i(b) = \mathcal{R}(r_a^i(b))$ to $\mathcal{S}$          // randomizer
9:             If total number of pulls reaches $T$, **exit**
10:         **end for**
11:         Send messages $\widehat{y}_a(b) = \mathcal{S}(\{y_a^i(b)\}_{1 \leq i \leq l(b)})$ to $\mathcal{A}$       // secure computation
12:         Compute the sum of rewards $R_a(b) = \mathcal{A}(\widehat{y}_a(b))$        // analyzer
13:         Compute mean estimate $\widehat{\mu}_a(b) = R_a(b)/l(b)$
14:         Compute confidence bounds $\text{UCB}_a(b) = \widehat{\mu}_a(b) + \beta(b)$ and $\text{LCB}_a(b) = \widehat{\mu}_a(b) - \beta(b)$
15:     **end for**
16:     Update active set of arms: $\Phi(b+1) = \{a \in \Phi(b) : \text{UCB}_a(b) \geq \max_{a' \in \Phi(b)} \text{LCB}_{a'}(b)\}$
17: **end for**
18: Subroutine: Local Randomizer $\mathcal{R}$ (**Input:** $x_i \in [0,1]$, **Output:** $y_i$)
19: **Require:** precision $g \in \mathbb{N}$, modulo $m \in \mathbb{N}$, batch size $n \in \mathbb{N}$, privacy level $\varepsilon$
20: Encode $x_i$ as $\widehat{x}_i = \lfloor x_i g \rfloor + \mathbf{Ber}(x_i g - \lfloor x_i g \rfloor)$
21: Generate discrete noise $\eta_i$ (depending on $n, \varepsilon, g$)         // random noise generator
22: Add noise and modulo clip $y_i = (\widehat{x}_i + \eta_i) \bmod m$
23: Subroutine: Secure Aggregation $\mathcal{S}$ (**Input:** $y_1, \ldots, y_n$, **Output:** $\widehat{y}$)
24: **Require:** modulo $m \in \mathbb{N}$
25: Securely compute $\widehat{y} = (\sum_{i=1}^n y_i) \bmod m$         // black-box function
26: Subroutine: Analyzer $\mathcal{A}$ (**Input:** $\widehat{y}$, **Output:** $z$)
27: **Require:** precision $g \in \mathbb{N}$, modulo $m \in \mathbb{N}$, batch size $n \in \mathbb{N}$, accuracy level $\tau \in \mathbb{R}$
28: **if** $\widehat{y} > ng + \tau$ **then**
29:     set $z = (\widehat{y} - m)/g$         // correction for underflow
30: **else** set $z = \widehat{y}/g$
---

**Fact 1** (Infinite Divisibility of Discrete Laplace). *A random variable $X$ has a Pólya distribution with parameters $r > 0, \beta \in [0,1]$, denoted by **Pólya**$(r, \beta)$, if it has a probability mass function given by* [3]

$$\forall x \in \mathbb{N}, \quad \mathbb{P}[X = x] = \frac{\Gamma(x+r)}{x!\Gamma(r)} \beta^x (1-\beta)^r.$$

*Now, for any $n \in \mathbb{N}$, let $\{\gamma_i^+, \gamma_i^-\}_{i \in [n]}$ be $2n$ i.i.d samples from **Pólya**$(1/n, e^{-1/b})$, then the random variable $\sum_{i=1}^n (\gamma_i^+ - \gamma_i^-)$ is distributed as **Lap**$_{\mathbb{Z}}(b)$.*

Further, to analyze privacy and regret, we will rely on the following fact on discrete Laplace [26].

**Fact 2** (Discrete Laplace Mechanism). *Let $\Delta, \varepsilon > 0$. Let $q : \mathcal{D}^n \to \mathbb{Z}$ satisfy $|q(D) - q(D')| \leq \Delta$ for all $D, D'$ differing on a single user's data. Define a mechansim $M : \mathcal{D}^n \to \mathbb{Z}$ by $M(D) = q(D) + Y$, where $Y \sim$ **Lap**$_{\mathbb{Z}}(\Delta/\varepsilon)$. Then, $M$ satisfies $(\varepsilon, 0)$-DP. Moreover, for all $m \in \mathbb{N}$,*

$$\mathbb{P}[Y > m] = \mathbb{P}[Y < -m] = \frac{e^{-\frac{\varepsilon m}{\Delta}}}{e^{\frac{\varepsilon}{\Delta}} + 1}.$$

Armed with these facts, we are able to obtain the following main theorem, which shows that the same regret as in the central model is achieved under the distributed model via SecAgg.

**Theorem 1** (Pure-DP via SecAgg). *Fix any $\varepsilon > 0$. Let $\mathcal{P} = (\mathcal{R}, \mathcal{S}, \mathcal{A})$ be a protocol such that the noise in $\mathcal{R}$ is given by $\eta_i = \gamma_i^+ - \gamma_i^-$, where $\gamma_i^+, \gamma_i^- \sim^{i.i.d.}$ **Pólya**$(1/n, e^{-\varepsilon/g})$. For each batch $b \geq 1$, choose $n = l(b), g = \lceil \varepsilon\sqrt{n} \rceil, \tau = \lceil \frac{g}{\varepsilon} \log(2/p) \rceil, p = 1/T$ and $m = ng + 2\tau + 1$. Then, Algorithm 1 instantiated with protocol $\mathcal{P}$ and confidence radius $\beta(b) = O\left(\sqrt{\frac{\log(|\Phi(b)|b^2/p)}{2l(b)}} + \frac{2\log(|\Phi(b)|b^2/p)}{\varepsilon l(b)}\right)$,*

---
[3]One can sample from **Pólya** as follows. First, sample $\lambda \sim$ **Gamma**$(r, \beta/(1-\beta))$ and then use it to sample $X \sim$ **Poisson**$(\lambda)$, which is known to follow **Pólya**$(r, \beta)$ distribution [28].

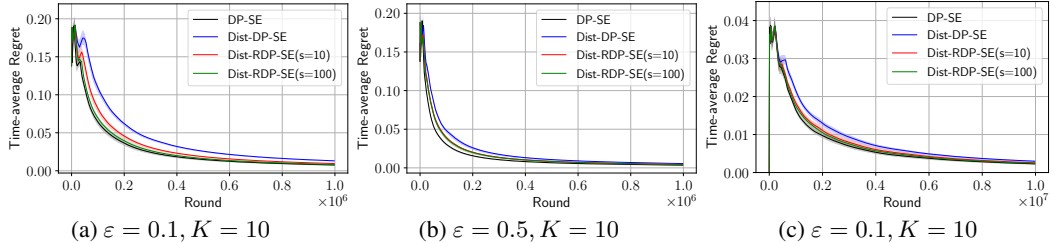

(a) $\varepsilon = 0.1, K = 10$      (b) $\varepsilon = 0.5, K = 10$      (c) $\varepsilon = 0.1, K = 10$

Figure 1: Comparison of time-average regret for Dist-DP-SE, Dist-RDP-SE, and DP-SE in Gaussian bandit instances under (a, b) large reward gap (easy instance) and (c) small reward gap (hard instance).

*achieves $(\varepsilon, 0)$-DP in the distributed model. Moreover, it enjoys the expected regret*

$$\mathbb{E}\left[Reg(T)\right] = O\left(\sum\nolimits_{a \in [K]:\Delta_a > 0} \frac{\log T}{\Delta_a} + \frac{K \log T}{\varepsilon}\right).$$

**Theorem 1 achieves optimal regret under pure DP.** Theorem 1 achieves the same regret bound as the one achieved in Sajed and Sheffet [11] under the central trust model with *continuous* Laplace noise. Moreover, it also matches the lower bound obtained under pure DP [9], indicating the bound is indeed tight Note that, we achieve this rate under distributed trust model – a stronger notion of privacy protection than the central model – while using only discrete privacy noise.

**Communication bits.** Algorithm 1 needs to communicate $O(\log m)$ bits per user to the secure protocol $\mathcal{S}$, i.e., communicating bits scales logarithmically with the batch size. In contrast, the number of communication bits required in existing distributed DP bandit algorithms that work with real-valued rewards (as we consider here) scale polynomially with the batch size [29, 30].

**MAB with distributed RDP guarantee.** In addition to pure DP, it is of natural interest to offer a slightly weaker privacy guarantee (but, still stronger than approximate DP) in the hope of gaining improvement in utility. One such notion of privacy is *Rényi differential privacy* (RDP) [31]. To this end, we obtain RDP in the distributed model by instantiating $\mathcal{R}$ at each user with a Skellam random noise. Proving a novel tail-bound for Skellam distribution, we show that a tighter regret bound compared to pure DP can be achieved (see Theorem 2)

## 5 Simulation Results

In this section, we empirically evaluate the regret performance of our successive elimination scheme with SecAgg protocol (Algorithm 1) under distributed trust model, which we abbreviate as Dist-DP-SE and Dist-RDP-SE when the randomizer $\mathcal{R}$ is instantiated with Pólya noise (achieves pure DP) and Skellam noise (achieves RDP), respectively. We compare them with the DP-SE algorithm of [11] that works only with *continuous* Laplace noise and achieves optimal regret under pure DP in the central model. We fix confidence level $p = 0.1$ and study comparative performances under varying privacy levels $\varepsilon < 1$. Similar to [32], we consider easy and hard MAB instances: in the former, arm means are sampled uniformly in $[0.25, 0.75]$, while in the latter, those are sampled in $[0.45, 0.55]$. We consider real rewards – sampled from Gaussian distribution with aforementioned means and projected to $[0, 1]$. We plot time-average regret $\text{Reg}(T)/T$ in Figure 1 by averaging results over 20 randomly generated bandit instances. We observe that as $T$ becomes large, the regret performance of Dist-DP-SE matches the regret of DP-SE. The slight gap in small $T$ regime is the cost that we pay to achieve distributed privacy using discrete noise without access to a trusted server (for higher $\varepsilon$ value, this gap is even smaller). In addition, we find that a relatively small scaling factor ($s = 10$) provides a considerable gain in regret under RDP compared to pure DP, especially when $\varepsilon$ is small (i.e., when the cost of privacy is not dominated by the non-private part of regret). The experimental findings are consistent with our theoretical results. Here, we note that our simulations are proof-of-concept only and we did not tune any hyperparameters. More details and additional plots are given in Appendix E.

## 6 Conclusion

We show that MAB under distributed trust model can achieve pure DP while maintaining the same regret under central model. In addition, RDP is also achieved in MAB udner distributed trust model for the first time. Both results are obtained via a unified algorithm design and performance analysis. More importantly, our work also opens the door to a promising research direction – private online learning with distributed DP guarantees, including contextual bandits and reinforcement learning.

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

# A  A General Regret Bound of Algorithm 1

In this section, we first present a generic regret bound for private MAB, see Lemma 1 below.

Recall that $n_a(b) := R_a(b) - \sum_{i=1}^{l(b)} r_a^i(b)$ is the total noise injected in the sum of rewards for arm $a$ during batch $b$. We consider the following tail property on $n_a(b)$.

**Assumption 1** (Concentration of Private Noise). *Fix any $p \in (0,1]$, $a \in [K]$, $b \geq 1$, there exist non-negative constants $\sigma, h$ (possibly depending on $b$) such that, with probability at least $1 - 2p$,*

$$|n_a(b)| \leq \sigma \sqrt{\log(2/p)} + h \log(2/p).$$

We remark that this assumption naturally holds for a single $(\sigma^2, h)$-sub-exponential noise and a single $\sigma^2$-sub-Gaussian noise where $h = 0$ (cf. Lemma 4 and Lemma 3 in Appendix D) with constants adjustment.

**Lemma 1.** *Let Assumption 1 hold and choose confidence radius*

$$\beta(b) = \sqrt{\frac{\log(4|\Phi(b)|b^2/p)}{2l(b)}} + \frac{\sigma\sqrt{\log(2|\Phi(b)|b^2/p)}}{l(b)} + \frac{h\log(2|\Phi(b)|b^2/p)}{l(b)}, \tag{1}$$

*where $|\Phi(b)|$ is the number of active arms in batch $b$. Then, for any $p \in (0,1]$, with probability at least $1 - 3p$, the regret of Algorithm 1 satisfies*

$$Reg(T) = O\left(\sum_{a \in [K]:} \frac{\log(KT/p)}{\Delta_a} + K\sigma\sqrt{\log(KT/p)} + Kh\log(KT/p)\right).$$

*Taking $p = 1/T$ and assuming $T \geq K$, yields the expected regret*

$$\mathbb{E}\left[Reg(T)\right] = O\left(\sum_{a \in [K]:\Delta > 0} \frac{\log T}{\Delta_a} + K\sigma\sqrt{\log T} + Kh\log T\right).$$

*Proof.* Let $\mathcal{E}_b$ be the event that for all active arms $|\widehat{\mu}_a(b) - \mu_a| \leq \beta(b)$ and $\mathcal{E} = \cup_{b \geq 1}\mathcal{E}_b$. Then, we first show that with the choice of $\beta(b)$ given by (1), we have $\mathbb{P}\left[\mathcal{E}\right] \geq 1 - 3p$ for any $p \in (0,1]$ under Assumption 1. To see this, we note that

$$\widehat{\mu}_a(b) - \mu_a = \frac{n_a(b) + \sum_{i=1}^{l(b)} r_a^i(b)}{l(b)} - \mu_a.$$

By Hoeffeding's inequality (cf. Lemma 5), we have for any $p \in (0,1)$, with probability at least $1 - p$,

$$\left|\frac{\sum_{i=1}^{l(b)} r_a^i(b)}{l(b)} - \mu_a\right| = \sqrt{\frac{\log(2/p)}{2l(b)}}.$$

Then, by the concentration of noise $n_a(b)$ in Assumption 1 and triangle inequality, we obtain for a given arm $a$ and batch $b$, with probability at least $1 - 3p$

$$|\widehat{\mu}_a(b) - \mu_a| = \sqrt{\frac{\log(2/p)}{2l(b)}} + \frac{\sigma\sqrt{\log(1/p)}}{l(b)} + \frac{h\log(1/p)}{l(b)}.$$

Thus, by the choice of $\beta(b)$ and a union bound, we have $\mathbb{P}\left[\mathcal{E}\right] \geq 1 - 3p$.

In the following, we condition on the good event $\mathcal{E}$. We first show that the optimal arm $a^*$ will always be active. We show this by contradiction. Suppose at the end of some batch $b$, $a^*$ will be eliminated, i.e., $\text{UCB}_{a^*}(b) < \text{LCB}_{a'}(b)$ for some $a'$. This implies that under good event $\mathcal{E}$

$$\mu_{a^*} \leq \widehat{\mu}_{a^*}(b) + \beta(b) < \widehat{\mu}_{a'}(b) - \beta(b) \leq \mu_{a'},$$

which contradicts the fact that $a^*$ is the optimal arm.

Then, we show that at the end of batch $b$, all arms such that $\Delta_a > 4\beta(b)$ will be eliminated. To show this, we have that under good event $\mathcal{E}$

$$\widehat{\mu}_a(b) + \beta(b) \leq \mu_a(b) + 2\beta(b) < \mu_{a^*}(b) - 4\beta(b) + 2\beta(b) \leq \widehat{\mu}_{a^*}(b) - \beta(b),$$

which implies that arm $a$ will be eliminated by the rule. Thus, for each sub-optimal arm $a$, let $\tilde{b}_a$ be the last batch that arm $a$ is not eliminated. By the above result, we have

$$\Delta_a \leq 4\beta(\tilde{b}_a) = O\left(\sqrt{\frac{\log(KT/p)}{l(\tilde{b}_a)}} + \frac{\sigma\sqrt{\log(KT/p)}}{l(\tilde{b}_a)} + \frac{h\log(KT/p)}{l(\tilde{b}_a)}\right).$$

Hence, we have for some absolute constants $c_1, c_2, c_3$,

$$l(\tilde{b}_a) \leq \max\left\{\frac{c_1\log(KT/p)}{\Delta_a^2}, \frac{c_2\sigma\sqrt{\log(KT/p)}}{\Delta_a}, \frac{c_3 h\log(KT/p)}{\Delta_a}\right\}$$

Since the batch size doubles, we have $N_a(T) \leq 4l(\tilde{b}_a)$ for each sub-optimal arm $a$. Therefore, $\text{Reg}(T) = \sum_{a \in [K]} N_a(T)\Delta_a \leq 4l(\tilde{b}_a)\Delta_a$. Moreover, choose $p = 1/T$ and assume $T \geq K$, we have that the expected regret satisfies

$$\begin{aligned}
\text{Reg}(T) &= \mathbb{E}\left[\sum_{a \in [K]} \Delta_a N_a(T)\right] \\
&\leq \mathbb{P}\left[\bar{\mathcal{E}}\right] \cdot T + O\left(\sum_{a \in [K]: \Delta > 0} \frac{\log T}{\Delta_a}\right) + O\left(K\sigma\sqrt{\log T}\right) + O\left(Kh\log T\right) \\
&= O\left(\sum_{a \in [K]: \Delta > 0} \frac{\log T}{\Delta_a} + K\sigma\sqrt{\log T} + Kh\log T\right).
\end{aligned}$$

$\square$

**Remark 1.** *In stead of a doubling batch schedule, one can also set $l(b) = \eta^b$ for some absolute constant $\eta > 1$ while attaining the same order of regret bound.*

# B   Appendix for Pure DP in Section 4

In this section, we provide proofs for Theorem 1, which show that pure DP can be achieved under the distributed model via SecAgg. The result builds on the generic regret bound in Lemma 1.

## B.1   Proof of Theorem 1

*Proof.* Privacy: We need to show that the server's view at each batch has already satisfies $(\varepsilon, 0)$-DP, which combined with the fact of unique users and parallel composition, yields that Algorithm 1 satisfies $(\varepsilon, 0)$-DP in the distributed model. To this end, in the following, we fix a batch $b$ and arm $a$, and hence $x_i = r_a^i(b)$ and $n = l(b)$. Note that the server's view for each batch is given by

$$\widehat{y} \overset{(a)}{=} \left(\sum_{i \in [n]} y_i\right) \bmod m \overset{(b)}{=} \left(\sum_{i \in [n]} \widehat{x}_i + \eta_i\right) \bmod m, \tag{2}$$

where (a) holds by SecAgg function; (b) holds by the distributive property: $(a + b) \bmod c = (a \bmod c + b \bmod c) \bmod c$ for any $a, b, c \in \mathbb{Z}$. Thus, the view of the server can be simulated as a post-processing of a mechanism $\mathcal{H}$ that accepts an input dataset $\{\widehat{x}_i\}_i$ and outputs $\sum_i \widehat{x}_i + \sum_i \eta_i$. Hence, it suffices to show that $\mathcal{H}$ is $(\varepsilon, 0)$-DP by post-processing of DP. To this end, we note that the sensitivity of $\sum_i \widehat{x}_i$ is $g$, which, by Fact 2, implies that $\sum_i \eta_i$ needs to be distributed as $\textbf{Lap}_{\mathbb{Z}}(g/\varepsilon)$ in order to guarantee $\varepsilon$-DP. Finally, by Fact 1, it suffices to generate $\eta_i = \gamma_i^+ - \gamma_i^-$, where $\gamma_i^+$ and $\gamma_i^-$ are i.i.d samples from $\textbf{Pólya}(1/n, e^{-\varepsilon/g})$.

Regret: Thanks to the generic regret bound in Lemma 1, we only need to verify Assumption 1. To this end, fix any batch $b$ and arm $a$, we have $\widehat{y} = \widehat{y}_a(b)$, $x_i = r_a^i(b)$ and $n = l(b)$. Then, in the following we will show that with probability at least $1 - 2p$

$$\left|\mathcal{A}(\widehat{y}) - \sum_i x_i\right| \leq O\left(\frac{1}{\varepsilon}\sqrt{\log(1/p)} + \frac{1}{\varepsilon}\log(1/p)\right), \tag{3}$$

which implies that Assumption 1 holds with $\sigma = O(1/\varepsilon)$ and $h = O(1/\varepsilon)$.

Inspired by [25, 24], we first divide the LHS of (3) as follows.

$$\left| \mathcal{A}(\widehat{y}) - \sum_i x_i \right| \le \underbrace{\left| \mathcal{A}(\widehat{y}) - \frac{1}{g} \sum_i \widehat{x}_i \right|}_{\text{Term (i)}} + \underbrace{\left| \frac{1}{g} \sum_i \widehat{x}_i - \sum_i x_i \right|}_{\text{Term (ii)}},$$

where Term (i) captures the error due to private noise and modular operation, and Term (ii) captures the error due to random rounding.

To start with, we will bound Term (ii). More specifically, we will show that for any $p \in (0, 1]$, with probability at least $1 - p$,

$$\text{Term (ii)} \le O\left( \frac{1}{\varepsilon} \sqrt{\log(1/p)} \right). \tag{4}$$

Let $\bar{x}_i := \lfloor x_i \cdot g \rfloor$, then $\widehat{x}_i = \bar{x}_i + \mathbf{Ber}(x_i g - \bar{x}_i) = x_i g + \bar{x}_i + \mathbf{Ber}(x_i g - \bar{x}_i) - x_i g = x_i g + \iota_i$, where $\iota_i := \bar{x}_i + \mathbf{Ber}(x_i g - \bar{x}_i) - x_i g$. We have $\mathbb{E}[\iota_i] = 0$ and $\iota_i \in [-1, 1]$. Hence, $\iota_i$ is 1-sub-Gaussian and as a result, $\frac{1}{g} \sum_i \widehat{x}_i - \sum_i x_i = \frac{1}{g} \sum_i \iota_i$ is $n/g^2$-sub-Gaussian. Therefore, by the concentration of sub-Gaussian (cf. Lemma 3), we have

$$\mathbb{P}\left[ \left| \sum_i \frac{1}{g} \cdot \widehat{x}_i - \sum_i x_i \right| > \sqrt{2 \frac{n}{g^2} \log(2/p)} \right] \le p.$$

Hence, by the choice of $g = \lceil \varepsilon \sqrt{n} \rceil$, we establish (4).

Now, we turn to bound Term (i). Recall the choice of parameters: $g = \lceil \varepsilon \sqrt{n} \rceil$, $\tau = \lceil \frac{g}{\varepsilon} \log(2/p) \rceil$, and $m = ng + 2\tau + 1$. We would like to show that

$$\mathbb{P}\left[ \left| \mathcal{A}(\widehat{y}) - \frac{1}{g} \sum_i \widehat{x}_i \right| > \frac{\tau}{g} \right] \le p, \tag{5}$$

which implies that for any $p \in (0, 1]$, with probability at least $1 - p$

$$\text{Term (i)} \le O\left( \frac{1}{\varepsilon} \log(2/p) \right). \tag{6}$$

To show (5), the key is to bound the error due to private noise and handle the possible underflow carefully. First, we know that the total private noise $\sum_i \eta_i$ is distributed as $\mathbf{Lap}_{\mathbb{Z}}(g/\varepsilon)$. Hence, by the concentration of discrete Laplace (cf. Fact 2), we have

$$\mathbb{P}\left[ \left| \sum_i \eta_i \right| > \tau \right] \le p.$$

Let $\mathcal{E}_{\text{noise}}$ denote the event that $\sum_i \widehat{x}_i + \eta_i \in [\sum_i \widehat{x}_i - \tau, \sum_i \widehat{x}_i + \tau]$, then by the above inequality, we have $\mathbb{P}[\mathcal{E}_{\text{noise}}] \ge 1 - p$. In the following, we condition on the event of $\mathcal{E}_{\text{noise}}$ to analyze the output $\mathcal{A}(\widehat{y})$. As already shown in (2), the input $\widehat{y} = (\sum_i \widehat{x}_i + \eta_i) \mod m$ is already an integer and hence $y = \widehat{y}$. We will consider two cases of $y$.

**Case 1:** $y > ng + \tau$. We argue that this happens only when $\sum_i \widehat{x}_i + \eta_i \in [-\tau, 0)$, i.e., underflow. This is because for all $i \in [n]$, $\widehat{x}_i \in [0, g]$, $m = ng + 2\tau + 1$ and the total privacy noise is at most $\tau$ under $\mathcal{E}_{\text{noise}}$. Therefore,

$$y - m = \left( \left( \sum_i \widehat{x}_i + \eta_i \right) \mod m \right) - m$$

$$= \left( m + \sum_i \widehat{x}_i + \eta_i \right) - m$$

$$= \sum_i \widehat{x}_i + \eta_i.$$

That is, $y - m \in [\sum_i \widehat{x}_i - \tau, \sum_i \widehat{x}_i + \tau]$ with high probability. In other words, we have show that when $y > ng + \tau$, $\mathcal{A}(\widehat{y}) = \frac{y-m}{g}$ satisfies (5).

**Case 2:** $y \le ng + \tau$. Here, we have noisy sum $\sum_i \widehat{x}_i + \eta_i \in [0, ng + \tau]$. Hence, $y = \sum_i \widehat{x}_i + \eta_i$ since $m = ng + 2\tau + 1$, which implies that $\mathcal{A}(\widehat{y}) = \frac{y}{g}$ satisfies (5).

Hence, we have shown that the output of the analyzer under both cases satisfies (5), which implies (6). Combined with the bound in (4), yields the bound in (3). Finally, plugging in $\sigma = O(1/\varepsilon)$, $h = O(1/\varepsilon)$ into the generic regret bound in Lemma 1, yields the required regret bound and completes the proof. $\qquad\square$

## C  Appendix for RDP

In the section, we will present the details of achieving Rényi differential privacy (RDP) [31] in the distributed DP model.

**Definition 4** (Rényi Differential Privacy). *For $\alpha > 1$, a randomized mechanism $\mathcal{M}$ satisfies $(\alpha, \varepsilon(\alpha))$-RDP if for all neighboring datasets $D, D'$, we have $D_\alpha(\mathcal{M}(D), \mathcal{M}(D')) \leq \varepsilon(\alpha)$, where $D_\alpha(P, Q)$ is the Rényi divergence (of order $\alpha$) of the distribution $P$ from the distribution $Q$, and is given by*

$$D_\alpha(P, Q) := \frac{1}{\alpha - 1} \log \left( \mathbb{E}_{x \sim Q} \left[ \left( \frac{P(x)}{Q(x)} \right)^\alpha \right] \right).$$

To achieve RDP guarantee using discrete noise, instead of discrete Laplace distribution in the previous section, we consider the Skellam distribution – which has recently been applied in federated learning [16]. In the multi-armed bandit setting, a key challenge in the regret analysis is to characterize the tail property of Skellam distribution. This is different from [16], where characterizing the variance of Skellam distribution is sufficient. In Proposition 1, we prove that Skellam has sub-exponential tails, which not only is the key to our regret analysis, but also could be of independent interest.

**Definition 5** (Skellam Distribution). *A random variable $X$ has a Skellam distribution with mean $\mu$ and variance $\sigma^2$, denoted by $\mathbf{Sk}(\mu, \sigma^2)$, if it has a probability mass function given by*

$$\forall x \in \mathbb{Z}, \quad \mathbb{P}[X = x] = e^{-\sigma^2} I_{x-\mu}(\sigma^2),$$

*where $I_\nu(\cdot)$ is the modified Bessel function of the first kind.*

To sample from Skellam distribution, one can rely on existing procedures for Poisson samples. This is because if $X = N_1 - N_2$, where $N_1, N_2 \sim^{\text{i.i.d.}} \mathbf{Poisson}(\sigma^2/2)$, then $X$ is $\mathbf{Sk}(0, \sigma^2)$ distributed. Moreover, due to this fact, Skellam is closed under summation, i.e., if $X_1 \sim \mathbf{Sk}(\mu_1, \sigma_1^2)$ and $X_2 \sim \mathbf{Sk}(\mu_2, \sigma_2^2)$, then $X_1 + X_2 \sim \mathbf{Sk}(\mu_1 + \mu_2, \sigma_1^2 + \sigma_2^2)$.

**Proposition 1** (Sub-exponential Tail of Skellam). *Let $X \sim \mathbf{Sk}(0, \sigma^2)$. Then, $X$ is $(2\sigma^2, \frac{\sqrt{2}}{2})$-sub-exponential. Hence, for any $p \in (0, 1]$, with probability at least $1 - p$,*

$$|X| \leq 2\sigma \sqrt{\log(2/p)} + \sqrt{2} \log(2/p).$$

With the above result, we can establish the following privacy and regret guarantee of Algorithm 1.

**Theorem 2** (RDP via SecAgg). *Fix any $\varepsilon > 0$. Let $\mathcal{P} = (\mathcal{R}, \mathcal{S}, \mathcal{A})$ be a protocol such that the noise in $\mathcal{R}$ is given by $\eta_i \sim \mathbf{Sk}(0, \frac{g^2}{n\varepsilon^2})$. Fix a scaling factor $s \geq 1$. For each batch $b \geq 1$, choose $n = l(b), g = \lceil s\varepsilon\sqrt{n} \rceil, \tau = \lceil \frac{2g}{\varepsilon} \sqrt{\log(2/p)} + \sqrt{2} \log(2/p) \rceil, p = 1/T$ and $m = ng + 2\tau + 1$. Then, Algorithm 1 instantiated with protocol $\mathcal{P}$ and confidence radius $\beta(b) = O\left( \sqrt{\frac{\log(|\Phi(b)|b^2/p)}{2l(b)}} + \frac{(1+1/s) \log(|\Phi(b)|b^2/p)}{\varepsilon l(b)} \right)$, achieves $(\alpha, \widehat{\varepsilon}(\alpha))$-RDP in the distributed model for all $\alpha = 2, 3, \ldots$, with $\widehat{\varepsilon}(\alpha) = \frac{\alpha\varepsilon^2}{2} + \min\left\{ \frac{(2\alpha-1)\varepsilon^2}{4s^2} + \frac{3\varepsilon}{2s^3}, \frac{3\varepsilon^2}{2s} \right\}$. Moreover, it enjoys the regret bound*

$$\mathbb{E}[Reg(T)] = O\left( \sum_{a \in [K]:\Delta_a > 0} \frac{\log T}{\Delta_a} + \frac{K\sqrt{\log T}}{\varepsilon} + \frac{K \log T}{s\varepsilon} \right).$$

**Privacy-Regret-Communication Trade-off.** Observe that the scaling factor $s$ allows us to achieve different trade-offs. If $s$ increases, both privacy and regret performances improve. In fact, for a sufficiently large value of $s$, the third term in the regret bound becomes sufficiently small, and we obtain an improved regret bound compared to Theorem 1. Moreover, the RDP privacy guarantee improves to $\widehat{\varepsilon}(\alpha) \approx \frac{\alpha\varepsilon^2}{2}$, which is the standard RDP rate for Gaussian mechanism [31]. However, a larger $s$ leads to an increase of communicating bits per user, but only grows logarithmically, since Algorithm 1 needs to communicate $O(\log m)$ bits to the secure protocol $\mathcal{S}$.

**RDP to Approximate DP.** To shed more insight on Theorem 2, we convert our RDP guarantee to approximate DP for a sufficiently large $s$. It holds that under the setup of Theorem 2, for sufficiently

large $s$, one can achieve $(O(\varepsilon), \delta)$-DP with regret $O\left(\sum_{a\in[K]:\Delta_a>0} \frac{\log T}{\Delta_a} + \frac{K\sqrt{\log T \log(1/\delta)}}{\varepsilon}\right)$ (via Lemma 7). The implication of this conversion is three-fold. **First**, this regret bound is $O(\sqrt{\log T})$ factor tighter than that achieved by Tenenbaum et al. [22] using a shuffle protocol with same $(\varepsilon, \delta)$-DP guarantee. **Second**, it yields a better regret performance compared to the bound achieved under $(\varepsilon, 0)$-DP in Theorem 1 when the privacy budget $\delta > 1/T$. This observation is consistent with the fact that a weaker privacy guarantee typically warrants a better utility bound. **Third**, this conversion via RDP also yields a gain of $O(\sqrt{\log(1/\delta)})$ in the regret when dealing with privacy composition (e.g., when participating users across different batches are *not unique*) compared to [22] that can only rely on approximate DP. This results from the fact that RDP provides a tighter composition compared to approximate DP.

## C.1 Proof of Proposition 1

*Proof.* We first establish the following result.

**Claim 1.** *For all $\lambda \in \mathbb{R}$, we have*
$$\cosh(\lambda) \leq e^{\lambda^2/2}.$$

To show this, by the infinite product representation of the hyperbolic cosine function, we have
$$\cosh(\lambda) = \prod_{k=1}^{\infty}\left(1 + \frac{4\lambda^2}{\pi^2(2k-1)^2}\right) \overset{(a)}{\leq} \exp\left(\sum_{k=1}^{\infty}\frac{4\lambda^2}{\pi^2(2k-1)^2}\right) \overset{(b)}{=} \exp(\lambda^2/2),$$
where (a) holds by the fact that $1 + x \leq e^x$ for all $x \in \mathbb{R}$ and (b) follows from $\sum_{k=1}^{\infty}\frac{1}{(2k-1)^2} = \frac{\pi^2}{8}$.

Then, note that $X \sim \mathbf{Sk}(0, \sigma^2)$, then its moment generating function (MGF) is given by $\mathbb{E}\left[e^{\lambda X}\right] = \exp(\sigma^2(\cosh(\lambda)-1))$. Hence, by the above claim, we have $\mathbb{E}\left[e^{\lambda X}\right] \leq \exp(\sigma^2(e^{\lambda^2/2}-1))$. Further, note that $e^x - 1 \leq 2x$ for $x \in [0,1]$. Thus, for $|\lambda| \leq \sqrt{2}$, we have
$$\mathbb{E}\left[e^{\lambda X}\right] \leq e^{\lambda^2\sigma^2} = e^{\frac{\lambda^2 2\sigma^2}{2}}.$$

Hence, by the definition of sub-exponential random variable (cf. Lemma 4), $X$ is $(2\sigma^2, \frac{\sqrt{2}}{2})$-sub-exponential, which again by Lemma 4 implies the required concentration result, i.e., for any $p \in (0, 1]$, with probability at least $1 - p$,
$$|X| \leq 2\sigma\sqrt{\log(2/p)} + \sqrt{2}\log(2/p).$$
$\square$

## C.2 Proof of Theorem 2

We will leverage the following result in [16, Theorem 3.5 ] to prove privacy guarantee.

**Lemma 2.** *For $\alpha \in \mathbb{Z}$, $\alpha > 1$, let $X \sim \mathbf{Sk}(0, \sigma^2)$. Then, an algorithm $M$ that adds $X$ to a sensitivity-$\Delta$ query satisfies $(\alpha, \varepsilon(\alpha))$-RDP with $\varepsilon(\alpha)$ given by*
$$\varepsilon(\alpha) \leq \frac{\alpha\Delta^2}{2\sigma^2} + \min\left\{\frac{(2\alpha-1)\Delta^2 + 6\Delta}{4\sigma^4}, \frac{3\Delta}{2\sigma^2}\right\}.$$

*Proof of Theorem 2.* Privacy: As before, we fix any batch $b$ and arm $a$, for simplicity, we let $x_i = r_a^i(b)$ and $n = l(b)$. Then, it suffices to show that the mechanism $\mathcal{H}$ that accepts an input dataset $\{\widehat{x}_i\}_i$ and outputs $\sum_i \widehat{x}_i + \sum_i \eta_i$ is private. To this end, since each local randomizer $\mathcal{R}$ generates noise $\eta_i \sim \mathbf{Sk}(0, \frac{g^2}{n\varepsilon^2})$ and Skellam is closed under summation, we have that the total noise $\sum_i \eta_i \sim \mathbf{Sk}(0, \frac{g^2}{\varepsilon^2})$. Thus, by Lemma 2 with $\Delta = g$, we have that for each batch $b$ with $n = l(b)$ and $g = \lceil s\varepsilon\sqrt{n}\rceil$, Algorithm 1 is $(\alpha, \widehat{\varepsilon}_n(\alpha))$-RDP with $\widehat{\varepsilon}_n(\alpha)$ given by
$$\widehat{\varepsilon}_n(\alpha) = \frac{\alpha\varepsilon^2}{2} + \min\left\{\frac{(2\alpha-1)\varepsilon^2}{4s^2n} + \frac{3\varepsilon}{2s^3n^{3/2}}, \frac{3\varepsilon^2}{2s\sqrt{n}}\right\}.$$
Since $n = l(b) > 1$, we have that for all batches $b \geq 1$,
$$\widehat{\varepsilon}_n(\alpha) \leq \widehat{\varepsilon}(\alpha) := \frac{\alpha\varepsilon^2}{2} + \min\left\{\frac{(2\alpha-1)\varepsilon^2}{4s^2} + \frac{3\varepsilon}{2s^3}, \frac{3\varepsilon^2}{2s}\right\}.$$

Regret: We will establish the following high probability bound so that we can apply our generic regret bound in Lemma 1

$$\left| \mathcal{A}(\widehat{y}) - \sum_i x_i \right| \leq O\left( \frac{1}{\varepsilon}\sqrt{\log(1/p)} + \frac{1}{s\varepsilon}\log(2/p) \right), \tag{7}$$

where $\widehat{y} := \widehat{y}_a(b)$ for each batch $b$ and arm $a$.

We again divide the LHS of (7) into

$$\left| \mathcal{A}(\widehat{y}) - \sum_i x_i \right| \leq \underbrace{\left| \mathcal{A}(\widehat{y}) - \frac{1}{g}\sum_i \widehat{x}_i \right|}_{\text{Term (i)}} + \underbrace{\left| \frac{1}{g}\sum_i \widehat{x}_i - \sum_i x_i \right|}_{\text{Term (ii)}}.$$

In particular, Term (ii) can be bounded by using the same method before, i.e.,

$$\mathbb{P}\left[ \left| \sum_i \frac{1}{g}\cdot\widehat{x}_i - \sum_i x_i \right| > \sqrt{2\frac{n}{g^2}\log(2/p)} \right] \leq p.$$

Hence, by the choice of $g = \lceil s\varepsilon\sqrt{n} \rceil$, we establish that with high probability

$$\text{Term (ii)} \leq O\left( \frac{1}{\varepsilon\cdot s}\sqrt{\log(1/p)} \right).$$

For Term (i), as in the previous proof, the key is to show that

$$\mathbb{P}\left[ \left| \sum_{i\in[n]} \eta_i \right| > \tau \right] \leq p.$$

To this end, we will utilize our established result in Proposition 1. Note that the total noise $\sum_i \eta_i \sim$ $\mathbf{Sk}(0, \frac{g^2}{\varepsilon^2})$, and hence by Proposition 1 and the choice of $\tau = \lceil \frac{2g}{\varepsilon}\sqrt{\log(2/p)} + \sqrt{2}\log(2/p) \rceil$, we have the above result. Following previous proof, this result implies that with high probability

$$\text{Term (i)} \leq \frac{\tau}{g} \leq O\left( \frac{1}{\varepsilon}\sqrt{\log(1/p)} + \frac{1}{s\varepsilon}\log(1/p) \right).$$

Combining the bounds on Term (i) and Term (ii), we have that the private noise satisfies Assumption 1 with constants $\sigma = O(1/\varepsilon)$ and $h = O(\frac{1}{s\varepsilon})$. Hence, by the generic regret bound in Lemma 1, we have established the required result. $\qquad\square$

## D   Auxiliary Lemmas

In this section, we summarize some useful facts that have been used in the paper.

**Lemma 3** (Concentration of sub-Gaussian). *A mean-zero random variable $X$ is $\sigma^2$-sub-Gaussian if for all $\lambda \in \mathbb{R}$*

$$\mathbb{E}\left[ e^{\lambda X} \right] \leq \exp\left( \frac{\lambda^2\sigma^2}{2} \right).$$

*Then, it satisfies that for any $p \in (0,1]$, with probability at least $1 - p$,*

$$|X| \leq \sqrt{2}\sigma\sqrt{\log(2/p)}.$$

**Lemma 4** (Concentration of sub-exponential). *A mean-zero random variable $X$ is $(\sigma^2, h)$-sub-exponential if for $|\lambda| \leq 1/h$*

$$\mathbb{E}\left[ e^{\lambda X} \right] \leq \exp\left( \frac{\lambda^2\sigma^2}{2} \right).$$

*Then, we have*

$$\mathbb{P}\left[ |X| > t \right] \leq 2\exp\left( -\min\left\{ \frac{t^2}{2\sigma^2}, \frac{t}{2h} \right\} \right).$$

*Thus, it satisfies that for any $p \in (0,1]$, with probability at least $1 - p$,*

$$|X| \leq \sqrt{2}\sigma\sqrt{\log(2/p)} + 2h\log(2/p).$$

**Lemma 5** (Hoeffding's Inequality). *Let $X_1, \ldots, X_n$ be independent and identically distributed (i.i.d) random variables and $X_i \in [0,1]$ with probability one. Then, for any $p \in (0,1]$, with probability at least $1 - p$,*

$$\left| \frac{1}{n} \sum_{i=1}^{n} X_n - \mathbb{E}\left[X_1\right] \right| \leq \sqrt{\frac{\log(2/p)}{2n}}.$$

**Lemma 6** (Sum of sub-exponential). *Let $\{X_i\}_{i=1}^{n}$ be independent zero-mean $(\sigma_i^2, h_i)$-sub-exponential random variables. Then, $\sum_i X_i$ is $(\sum_i \sigma_i^2, h_*)$-sub-exponential, where $h_* := \max_i h_i$. Thus, we have*

$$\mathbb{P}\left[ \left| \sum_i X_i \right| > t \right] \leq 2 \exp\left( -\min\left\{ \frac{t^2}{2\sum_i \sigma_i^2}, \frac{t}{2h_*} \right\} \right).$$

*In other words, for any $p \in (0,1]$, if $v \geq \max\{\sqrt{2\sum_i \sigma_i^2 \log(2/p)}, 2h_* \log(2/p)\}$, with probability at least $1 - p$, $|\sum_i X_i| \leq v$.*

**Lemma 7** (Conversion Lemma). *If $\mathcal{M}$ satisfies $(\alpha, \varepsilon(\alpha))$-RDP, then for any $\delta \in (0,1)$, $\mathcal{M}$ satisfies $(\varepsilon, \delta)$-DP where*

$$\varepsilon = \inf_{\alpha > 1} \varepsilon(\alpha) + \frac{\log(1/(\alpha\delta))}{\alpha - 1} + \log(1 - 1/\alpha).$$

*If $\mathcal{M}$ satisfies $\frac{1}{2}\varepsilon^2$-CDP, then for for any $\delta \in (0,1)$, $\mathcal{M}$ satisfies $(\varepsilon', \delta)$-DP where*

$$\varepsilon' = \inf_{\alpha > 1} \frac{1}{2}\varepsilon^2 \alpha + \frac{\log(1/(\alpha\delta))}{\alpha - 1} + \log(1 - 1/\alpha) \leq \varepsilon \cdot \left( \sqrt{2\log(1/\delta)} + \varepsilon/2 \right).$$

*Moreover, if $\mathcal{M}$ satisfies $(\varepsilon, 0)$-DP, then it satisfies $(\alpha, \frac{1}{2}\varepsilon^2 \alpha)$-RDP simultaneously for all $\alpha \in (1, \infty)$.*

**Theorem 3** (Advanced composition). *Given target privacy parameters $\varepsilon' \in (0,1)$ and $\delta' > 0$, to ensure $(\varepsilon', k\delta + \delta')$-DP for the composition of $k$ (adaptive) mechanisms, it suffices that each mechanism is $(\varepsilon, \delta)$-DP with $\varepsilon = \frac{\varepsilon'}{2\sqrt{2k \log(1/\delta')}}$.*

## E  More Details on Simulations

We numerically compare the performance of Algorithm 1 under pure-DP and RDP guarantees in the distributed model (named Dist-DP-SE and Dist-RDP-SE, respectively) with the DP-SE algorithm of Sajed and Sheffet [11], which achieves pure-DP under the central model. We vary the privacy level as $\varepsilon \in \{0.1, 0.5, 1\}$, where a lower value of $\varepsilon$ indicates higher level of privacy.

In Figure 2, we consider the *easy* instance, i.e., where arm means are sampled uniformly in $[0.25, 0.75]$. In Figure 3, we consider the *hard* instance, i.e., where arm means are sampled uniformly in $[0.45, 0.55]$. The sampled rewards are Gaussian distributed with the given means and truncated to $[0,1]$. We plot results for $K = 10$ arms.

We see that, for higher value of time horizon $T$, the time-average regret of Dist-DP-SE is order-wise same to that of DP-SE, i.e., we are able to achieve similar regret performance in the distributed trust model as that is achieved in the central trust model. As mentioned before, we observe a gap for small value of $T$, which is the price we pay for discrete privacy noise (i.e., additional data quantization error on the order of $O(\frac{1}{\varepsilon}\sqrt{\log(1/p)})$) and not requiring a trusted central server. Hence, if we lower the level of privacy (i.e., higher value of $\varepsilon$), this gap becomes smaller, which indicates an inherent trade-off between privacy and utility.

We also observe that if we relax the requirement of privacy from pure-DP to RDP, then we can achieve a considerable gain in regret performance; more so when privacy level is high (i.e., $\varepsilon$ is small). This gain depends on the scaling factor $s$ – the higher the scale, the higher the gain in regret.

In Figure 4, we compare regret achieved by our generic batch-based successive arm elimination algorithm (Algorithm 1) instantiated with different protocols $\mathcal{P}$ under different trust models and privacy guarantees: (i) central model with pure-DP (CDP-SE), (ii) local model with pure-DP (LDP-SE), (iii) Distributed model with pure-DP (Dist-DP-SE), Renyi-DP (Dist-RDP-SE) and Concentrated-DP (Dist-CDP-SE). First, consider the pure-DP algorithms. We observe that regret performance of CDP-SE and Dist-DP-SE is similar (with a much better regret than LDP-SE). Now, if we relax the pure-DP requirement, then we achieve better regret performance both for Dist-RDP-SE and Dist-CDP-SE. Furthermore, Dist-CDP-SE performs better in terms of regret than Dist-RDP-SE. This

is due to the fact that under CDP, we use discrete Gaussian noise (which has sub-Gaussian tails) as opposed to the Skellam noise (which has sub-exponential tails) used under RDP.

**Experiment on Real Data**   We evaluate the performance of our algorithm on bandit instances generated from Microsoft Learning to Rank dataset MSLR-WEB10K [33]. The dataset consists of 1200192 rows and 138 columns, where each row corresponds to a query-url pair. The first column is relevance label 0, 1, ... , 4 of the pair, which we take as rewards. The second column denotes the query id, and the rest 136 columns denote contexts of a query-url pair. We cluster the data by running K-means algorithm with $K = 50$. We treat each cluster as a bandit arm with mean reward as the empirical mean of the individual ratings in the cluster. This way, we obtain a bandit setting with number of arms $K = 50$. We average over 10 parallel runs and plot the results in Figure 5 for privacy levels $\varepsilon \in \{1, 5, 10\}$. Similar to synthetic data experiments, we observe that as $T$ becomes large, the regret performance of our algorithm (Dist-DP-SE) under distributed model matches the regret of the state-of-the-art algorithm (DP-SE) under central model. Furthermore, consistent with observations on synthetic data, we also observe a considerable gain in regret under RDP compared to pure DP.

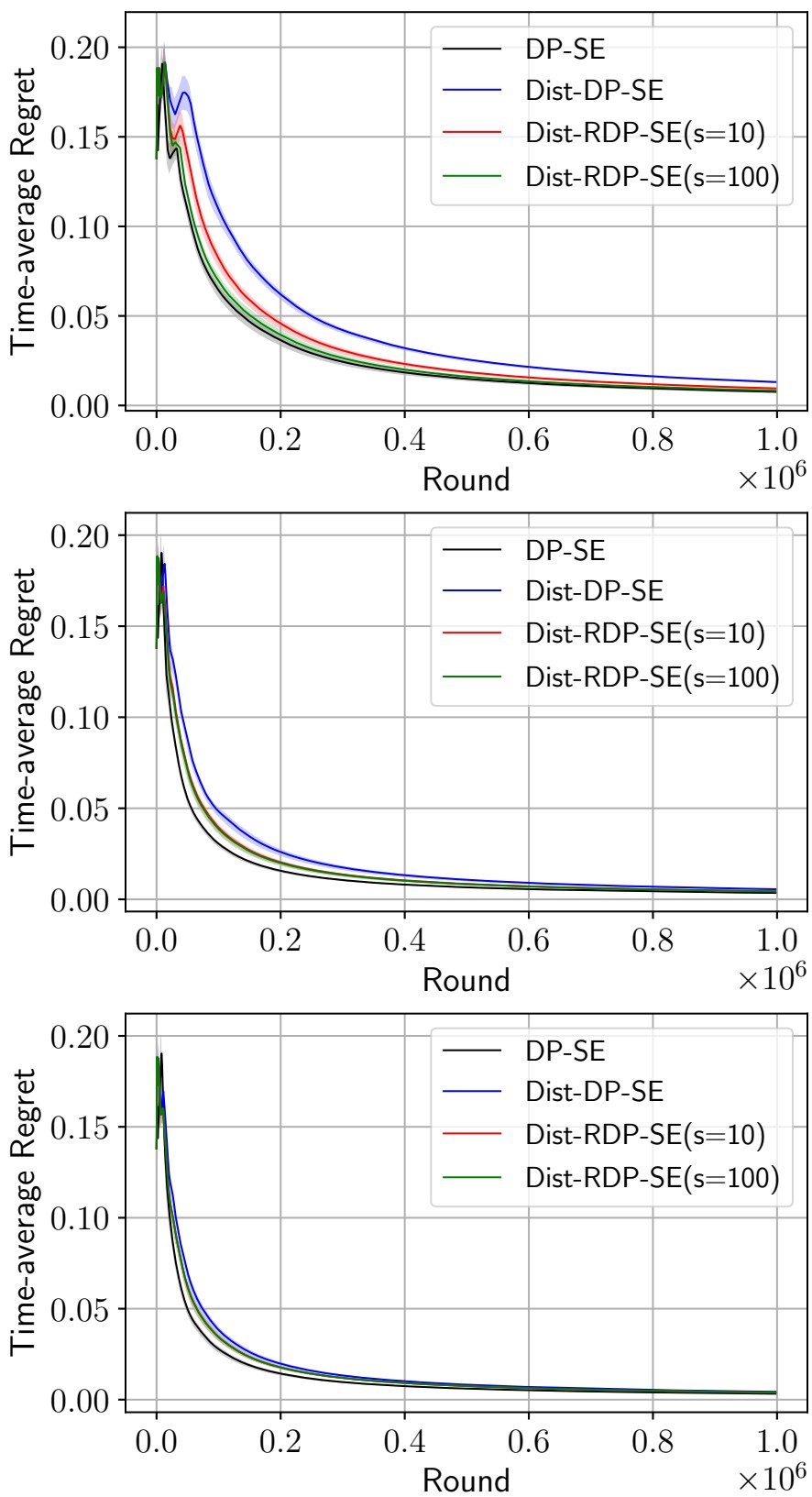

Figure 2: Comparison of time-average regret for Dist-DP-SE, Dist-RDP-SE, and DP-SE in Gaussian bandit instances under large reward gap (easy instance) with privacy level $\varepsilon = 0.1$ (top), $\varepsilon = 0.5$ (mid) and $\varepsilon = 1$ (bottom)

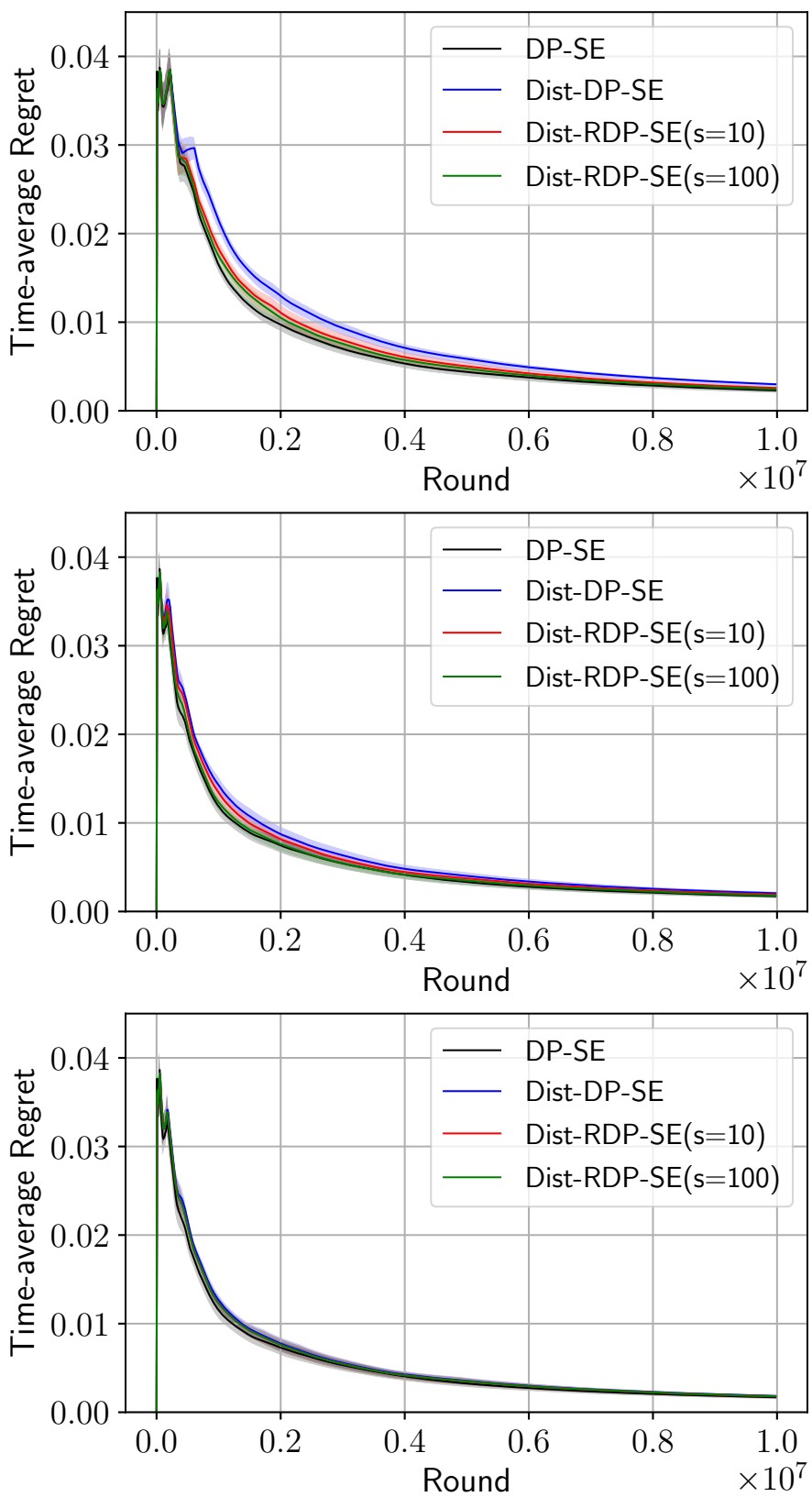

Figure 3: Comparison of time-average regret for Dist-DP-SE, Dist-RDP-SE, and DP-SE in Gaussian bandit instances under small reward gap (hard instance) with privacy level $\varepsilon = 0.1$ (top), $\varepsilon = 0.5$ (mid) and $\varepsilon = 1$ (bottom)

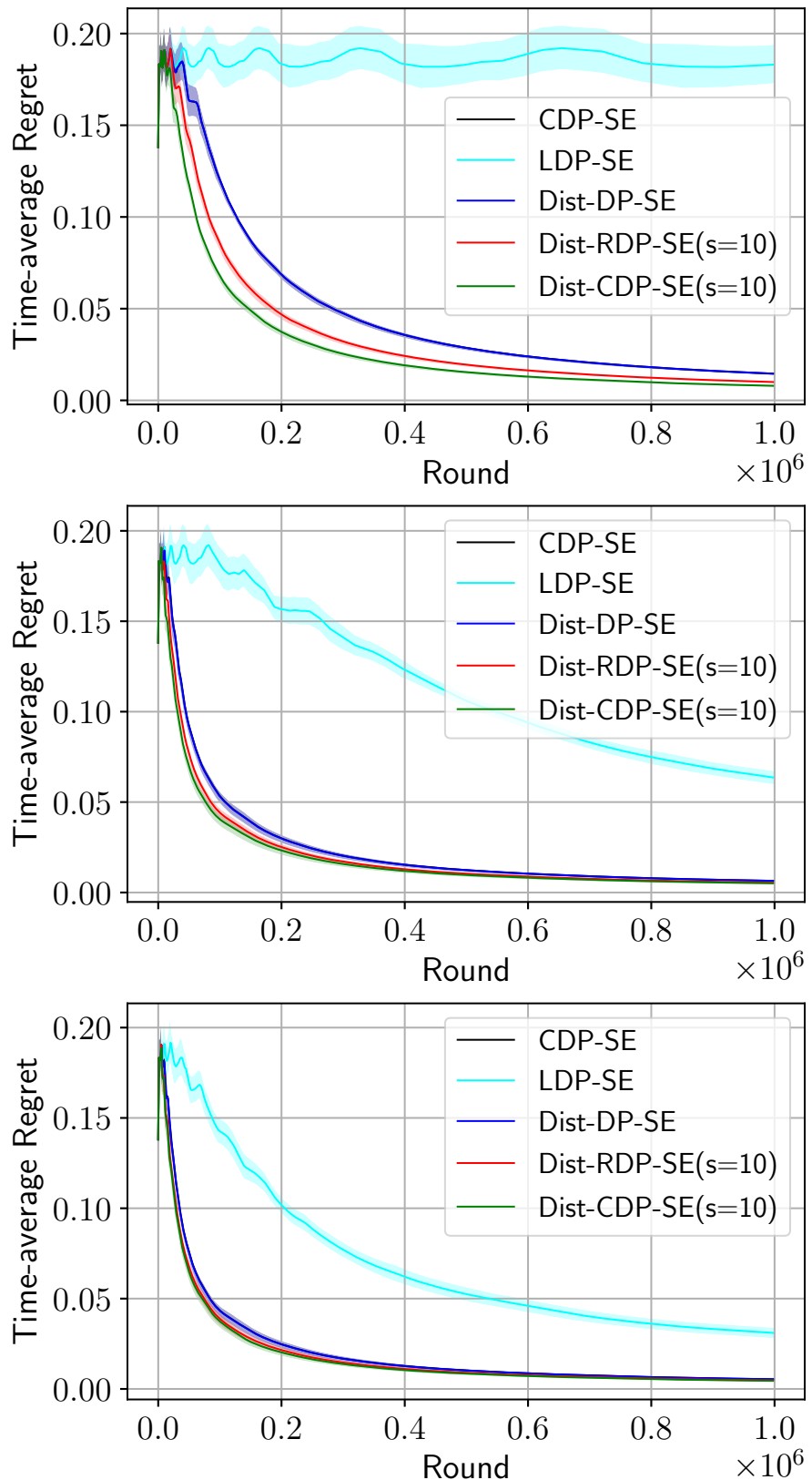

Figure 4: Comparison of time-average regret for CDP-SE, LDP-SE, Dist-DP-SE, Dist-RDP-SE and Dist-CDP-SE in Gaussian bandit instances under large reward gap (easy instance) with privacy level $\varepsilon = 0.1$ (top), $\varepsilon = 0.5$ (mid) and $\varepsilon = 1$ (bottom)

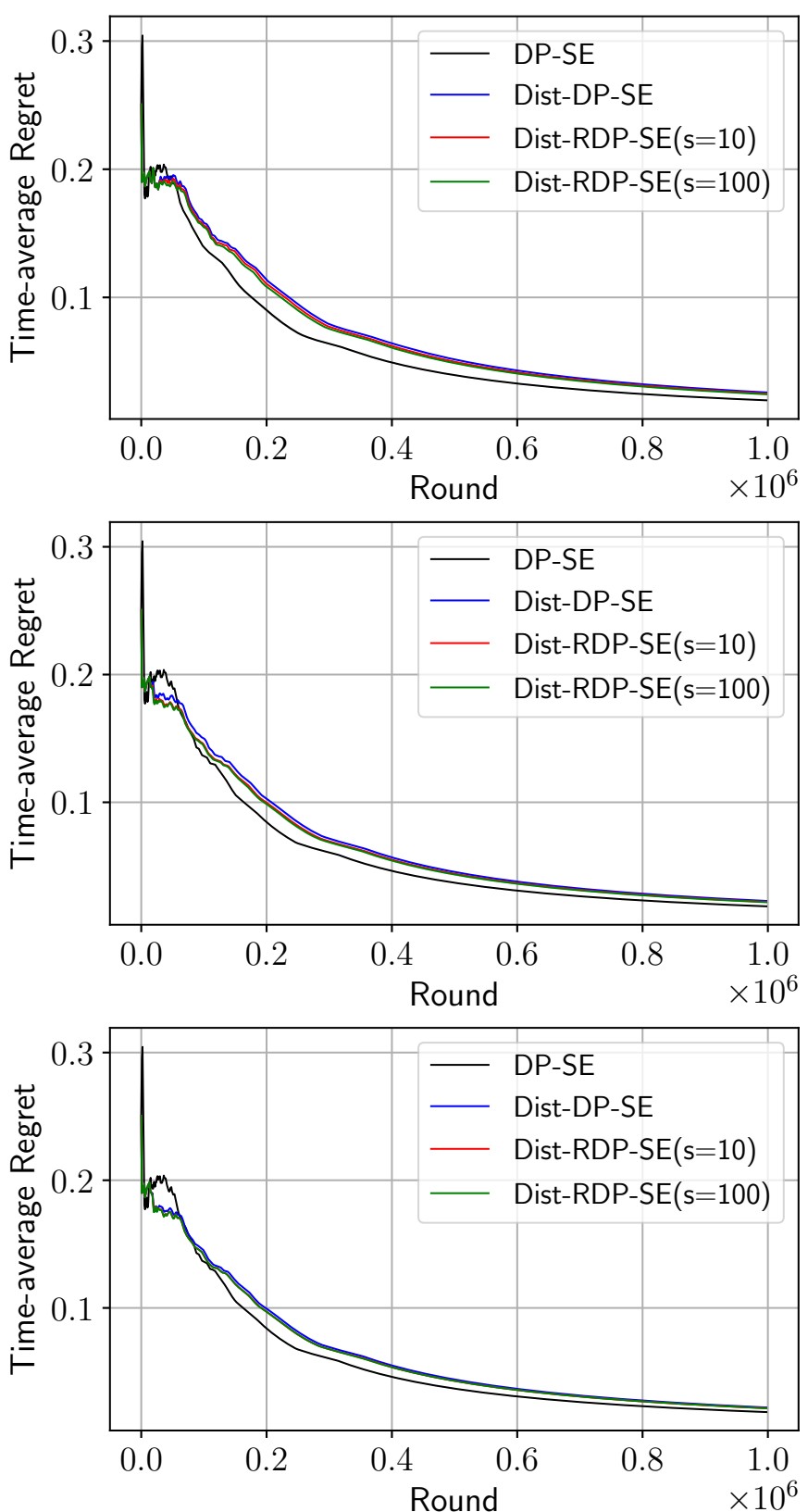

Figure 5: Comparison of time-average regret for Dist-DP-SE, Dist-RDP-SE, and DP-SE in bandit instances generated from real dat with privacy level $\varepsilon = 1$ (top), $\varepsilon = 5$ (mid) and $\varepsilon = 10$ (bottom)

