# OpenReview forum: "Distributed Differential Privacy in Multi-Armed Bandits"
_NeurIPS.cc/2022/Workshop/TSRML — TSRML2022_

### Official Review · Reviewer_Zxtn · 2022-10-19
**A practical algorithm for DP multiarmed bandits with untrusted server plus an analysis that improves the known utility bounds**

**Overall Rating:** 6

**Summary:**

I think this is a high-level DP paper that gives both a practical algorithm and a novel analysis to private training multi-armed bandits with untrusted central aggregator.

As mentioned in the paper, multi armed bandits (MABs) have several applications (online advertising, product recommendations, clinical trials etc.) and it is easy to think of a situation where we would like to have an algorithm that assumes an untrusted central server.

For aggregating users' information in this setting (untrusted central server) for training MAB, so called shufflers have been used before. Here users add noise to their messages (i.e. local DP messages) and central DP guarantees are obtained after aggregation. This paper proposes using secure aggregation instead of shuffling, and addresses practical issues that come with it (integer values messages etc., to which tailored noise adding mechanisms have to be used).

One benefit of this approach is that the central DP is also eps-DP. It is impressive that the paper provides a utility bound that is (asymptotically) lower than the (eps,delta)-bound of the baseline method.

**Strengths:**

- Clear contribution: improving the regret bound for MAB with a simple pure eps-DP approach that is based on the discrete Laplace mechanism
- Very well written, this seems like a solid paper. I didn't go through all the math but the parts I did read more carefully seem correct.
- I can imagine this is highly practical algorithm as well (applications mentioned: advertising, product recommendations etc.)


**Weaknesses:**

This is the only proper weakness coming to my mind but might be big one:

- Suitability for this venue. For this venue, one problem might be that this is so theoretical and niché that it might not open up to the audience. I can imagine this would be a perfect fit to a workshop/conference specializing in DP or theoretical CS.
- You don't discuss the possible benefits of using shuffling vs. secure aggregation. Wouldn't it be easier for the aggregator to see all the messages when using shuffling and  discard 'bad ones' (e.g. poisoning attacks etc.) from model updates ?


Remarks:

As far as I understand, you use RDP approach (included in Appendix) as well in Figure 1. And to get epsilons you need to convert the RDP values to (eps,delta).
What is delta? I cannot see it mentioned anywhere.



p.6, line 228: 'udner'


" it is of natural interest to offer a slightly weaker privacy guarantee (but, still stronger than approximate DP)"
I would not say that RDP nor approximate DP are weaker guarantees than pure DP.. pure-DP is (eps,0)-approximate DP having a
(eps,delta) curve instead of just single eps for pure eps-DP may simply give you more information about the privacy profile.
Notice that you need to have Laplace noise to be able to use pure eps-DP.
How come you think that RDP is stronger than approximate DP? I would claim that they give similar relaxation, however approximate DP is a more natural one.

I think there ought to be reference to [16] when you talk about the Skellam mechanism in the main text.



**Overall Recommendation:**

Despite of being quite DP specialized paper and quite theoretical, I would still recommend this for acceptance.

On the other hand I understand, if there is strict quota, that this gets rejected as it is possible it does not open up to a wider audience that this workshop might have. This paper would be a perfect fit to DP workshop or theoretical CS workshop.

**Review Confidence:**

3: The reviewer is fairly confident that the evaluation is correct

---

### Official Review · Reviewer_hW1x · 2022-10-21
**Interesting Paper**

**Overall Rating:** 7

**Summary:**

This paper proposed an algorithm for guaranteeing pure DP of Multi-Armed Bandits in the distributed trust model using Laplace noise. This work shows that they can obtain the optimal expected regret with both theory and simulation to support this. To relax to RDP, this work proves a tail bound on skellam noise.

**Strengths:**

1. The paper is clear and well-written.

2. Achieves clear improvement over continuous Laplace mechanisms by improving the trust model while maintaining the same expected regret.

3. Includes both theoretical and empirical simulations.

4. algorithm is clear and well-designed.

5. references appear comprehensive, though comparison with FL literature could be better.

**Weaknesses:**

1. Algorithm 1 appears very similar to the distributed DP algorithm from FL [16, 17], though tailored to MAB and using Laplacian noise. Comparing with these algorithms could be beneficial.

2. Though prior Distributed DP work in MAB required bits scaling polynomially in m, prior distributed DP work in FL showed that this only required communication scaling in O(log(m)) [16, 17], or less (see The Fundamental Price of Secure Aggregation in Differentially Private Federated Learning). It is unclear why this is not expected here. Note that in FL, the equivalent "rewards" (gradient values) are also real-valued.

Nits:

"the distributed DP" on line 122

"." Note on line 195


**Overall Recommendation:**

This paper provides a clear improvement over prior work in multi-armed bandits. Though the work only included simulations of the regret, the detailed theoretical analysis presents improvement in pure distributed DP for this setting. However, the paper could be benefitted by some more comparison to existing distributed DP literature in other domains.

**Review Confidence:**

3: The reviewer is fairly confident that the evaluation is correct

---

### Decision · Program_Chairs · 2022-10-23

**Decision:**

Accept

**Comment:**

Following the unanimous recommendations from reviewers, the submission is accepted.